# Effect of virtual reality (VR) technology on anxiety control and acrophobia reduction: A randomized controlled trial in Iran

## Research Article

automation; digital health; virtual reality; virtual reality exposure therapy; mental health; anxiety phobic disorders

**Corresponding author:**
Majid Zare-Bidaki;
Email: m.zare@live.co.uk

Cathal Breen[1], Freshteh Osmani[2], Seyed Reza Sajadi[3], Saeed Mohtasham[2], Mohsen Khosravi[2] ⬤ and Majid Zare-Bidaki[2]

[1]Edinburgh Napier University, UK; [2]Birjand University of Medical Sciences, Iran, Islamic Republic of and [3]Mashhad University of Medical Sciences, Iran, Islamic Republic of

## Abstract

In recent years, utilizing technologies, such as virtual reality in mental healthcare and treatment, has developed significantly. This study aimed to investigate the effect of using virtual reality (VR) technology on controlling anxiety and reducing fear of heights (acrophobia). This study was a randomized controlled trial conducted in Birjand, Iran, in 2020. 120 participants were recruited and randomly allocated into two groups: intervention and control. The intervention group underwent a single simulated exposure to height using a virtual reality headset. The Beck Anxiety Inventory, alongside a researcher-developed questionnaire were administered as pre-tests to assess acrophobia. Data analysis was performed using SPSS version 23, with significance level at 0.05. The intervention group showed significantly reduced anxiety and acrophobia scores immediately and 1 month after exposure (P < 0.05). Post-exposure, both anxiety and acrophobia scores were significantly lower in the intervention group compared to the control group (P = 0.03 and P < 0.001, respectively), with no significant differences between groups before exposure or 1 month later (P > 0.05). The study concluded that VR technology is an effective tool for reducing anxiety and acrophobia. This approach appears to hold significant promise as a therapeutic modality for psychiatrists treating patients with acrophobia.

## Impact Statements

The study findings confirmed the appropriateness of employing virtual reality as an effective intervention for reducing anxiety and acrophobia. These results offer mental health policy-makers and administrators essential evidence to support the informed integration of virtual reality into healthcare services, with careful consideration of its strengths and limitations for each specific disorder. Furthermore, the study enabled patients to make informed decisions regarding the use of virtual reality interventions in managing their conditions.

## Introduction

In the contemporary industrialized world, anxiety has been identified as a significant factor affecting mental health (Remes et al., 2016). Phobia, classified as an anxiety disorder, is characterized by a persistent and excessive fear of a specific object or situation lasting for more than 6 months (Eaton et al., 2018). A review of population-based studies worldwide reports a high prevalence of specific phobias, with median lifetime prevalence rates around 7.2% (Eaton et al., 2018). Individuals with specific phobias exhibit a consistent fear response either upon encountering the particular object or situation or in anticipation of such encounters (Clark and Rock, 2016; Osmani and Azarkar, 2021).

Acrophobia is an irrational fear of heights, which, according to the "Diagnostic and Statistical Manual of Mental Disorders (DSM-5)," is considered a specific phobia. DMS-5 provided six diagnostic criteria for acrophobia and other specific phobias as follows: (1) Unreasonable and excessive fear, (2) Immediate anxiety response, (3) Avoidance or extreme distress, (4) Life-limiting, (5) Six months duration, (6) Not caused by another disorder (Martin, 2003; Arroll et al., 2017). This chronic disorder can seriously affect people's lives and prevent them from being able to perform daily tasks, such as climbing stairs or standing on the balcony. It also can cause distress while performing recreational activities (Al-Agroudi et al., 2016). In such a context, acrophobic behavior usually involves avoiding various height-related conditions such as stairs, terraces, apartments and offices located in tall buildings, bridges, elevators and air travel (Arroll et al., 2017).

Various treatments have been proposed for specific phobias, including cognitive therapy, exposure therapy, behavioral therapy, systematic desensitization, hypnotherapy, supportive therapy, pharmacotherapy and exposure in virtual reality environments (Ayala et al., 2009;

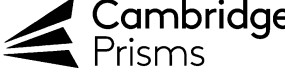



Osmani, 2020). Exposure therapy is a cognitive-behavioral method for the treatment of anxiety disorders (10). It is a multifaceted, developed method that targets behavior, cognition and emotions (Gromer et al., 2018). Exposure-based treatments are considered the gold standard in the treatment of specific phobias (Moore et al., 2002; Deacon and Abramowitz, 2004; Osmani et al., 2019). In such a context, utilizing virtual reality (VR) is one of the methods for exposure therapy that has been increasingly studied in patients with certain phobias in recent years (J. R. S. Freitas et al., 2021; Kuleli et al., 2025; Maples-Keller et al., 2017; Rimer et al., 2021; Wechsler et al., 2019). In a virtual environment, the condition in which phobia occurs is simulated as a three-dimensional (3D), 360-degree video, using a combination of graphics, visual displays, graphic games, body tracking devices and other sensory input devices (Maples-Keller et al., 2017).

The term VR was first introduced by Jaron Lanier about half a century ago, and nowadays it has effective and beneficial applications in various scientific areas (Botella et al., 2017). Its application has gained popularity among both psychologists and clients due to the potential of creating and designing a variety of therapeutic and educational environments, as many clients prefer to choose safe conditions to face their phobias, and VR can increase their motivation (Yuen et al., 2013). Virtual reality exposure therapy (VRET) has been promoted as a new tool that is both safe and cost-effective compared with real-life exposure (Freeman et al., 2017; Morina et al., 2021; Osmani, 2021).

Multiple reviews within the existing literature have examined the effects of virtual reality (VR) interventions on anxiety and phobic disorders. Notably, a systematic review and meta-analysis evaluated the comparative efficacy of virtual reality exposure therapy versus in vivo exposure therapy for social anxiety and specific phobia. The findings indicated that both treatment modalities are similarly effective in mitigating symptoms of social anxiety and phobia, with both approaches demonstrating moderate effect sizes (Kuleli et al., 2025). Another systematic review and meta-analysis evaluated the impact of virtual reality applications in the treatment of anxiety disorders. The analysis revealed a non-significant overall effect size accompanied by significant heterogeneity across studies. However, when compared to passive control groups, virtual reality interventions were significantly associated with reduced anxiety symptoms. This suggests that VR-based treatments may be beneficial in alleviating anxiety, although variability in study designs and populations indicates the need for further high-quality research (Schröder et al., 2023). Finally, another systematic review and meta-analysis comparing the relative efficacy of VRET and in vivo exposure therapy among individuals with phobias found generally positive outcomes for VRET across most phobias. However, for certain specific phobias, standard in vivo exposure demonstrated superior effectiveness. The results suggested that for some phobias, VRET may not achieve the same levels of immersion and presence as in vivo exposure, potentially impacting its efficacy in those cases (José Rúben Silva Freitas et al., 2021).

Although a substantial body of literature exists globally regarding the effects of virtual reality technology in reducing anxiety, particularly acrophobia, there is a limited number of studies conducted in low-resource settings. Consequently, research involving populations with limited exposure to flying and heights, such as those in Iran, remains scarce. In this regard, one study evaluated the efficacy of VRET in reducing anxiety symptoms among Iranian individuals suffering from flying phobia (aerophobia). The researchers implemented VRET as an intervention and assessed its impact on participants' anxiety levels related to flying. The results demonstrated that VRET significantly decreased anxiety symptoms, indicating that this therapeutic approach was effective in alleviating flying phobia in the studied population (Manshaee et al., 2020). Another similar study investigated the effectiveness of a VR program on behavioral functions, emotional regulation and brain functions in the treatment of aerophobia. Using a semi-experimental pre-test-post-test design with an experimental group receiving VR training and a control group, the researchers assessed 26 participants diagnosed with aerophobia in Tehran. The intervention consisted of five weekly VR sessions, each lasting 60 minutes. Data were collected through standardized questionnaires on fear of flying and emotion regulation, as well as functional near-infrared spectroscopy (fNIRS) to measure brain activity. The results demonstrated that the VR program significantly improved behavioral functions related to fear of flying (P < 0.01), enabling participants to better manage their fear and use air travel without distress. However, the intervention did not produce significant changes in emotional regulation or brain activity indicators (P > 0.01). The study concluded that VR-based treatment is effective in enhancing behavioral responses in individuals with aerophobia and can serve as a valuable therapeutic tool for this phobia (Lotfizadeh et al., 2025).

The assessment of the effects of VRET on various disorders, particularly mental health conditions where real-life interventions may be costly and carry potential risks for both healthcare providers and clients, holds significant implications for all stakeholders. In this context, healthcare providers can utilize such data to inform the evidence-based integration of VRET into healthcare services, carefully considering the strengths and limitations of this therapeutic approach for each specific disorder. Likewise, patients can evaluate the effectiveness of these interventions in managing their conditions. Accordingly, the present study aimed to investigate the impact of virtual reality technology on reducing fear and anxiety related to heights (acrophobia).

## Methods

This randomized controlled trial was conducted at Birjand University of Medical Sciences, located in Birjand city, in the eastern region of Iran. The study data were reported in accordance with the CONSORT (Consolidated Standards of Reporting Trials) 2025 guideline for randomized controlled trials (Hopewell et al., 2025).

### Data collection

Initially, the study participants were exposed to a simulated height experience using VR glasses. Subsequently, the Beck Anxiety Inventory (BAI) and a researcher-developed acrophobia questionnaire were administered as pre-test measures, following the formats established in previous studies (Ferrer-García et al., 2017; du Sert et al., 2018).

### Study participants

As presented in Figure 1, a total of 120 students from Birjand University of Medical Sciences were voluntarily recruited. This sample size was considered appropriate and sufficient, as the existing literature includes studies with sample sizes as small as 20 individuals, whereas samples exceeding 100 participants are rarely reported in published research, as indicated by previous systematic reviews (Savoric et al., 2025; Zeng et al., 2025). In such a context,

individuals who self-reported experiencing acrophobia were contacted and invited to participate in the study, contingent upon their agreement to join as research participants. Demographic characteristics of the study participants, including age, sex and level of education, were collected to enable a more comprehensive and precise analysis of the study results. The demographic data were carefully incorporated into a Microsoft Office 2020 file by one researcher, and the data were supervised and validated by another researcher.

Participant inclusion criteria: (1) The diagnosis of acrophobia was established according to the DSM-5 criteria, requiring the presence of symptoms for a minimum duration of 6 months, and confirmed through clinical evaluation by two qualified experts in the relevant field. (2) The absence of any other mental disorders.

Participant exclusion criteria: (1) pregnancy, (2) current drug or psychiatric treatment or within 3 months before the start of the study, (3) cardiovascular or neurological diseases, (4) stereoscopic visual impairment, (5) severe aphthous disorder, (6) severe psychiatric disorder, (7) dissatisfaction with continuing to be part of the study.

### Study intervention

The study participants were randomly allocated into two groups, intervention and control, using the block randomization technique. This technique ensures equal group sizes by dividing participants into blocks, thereby maintaining balance throughout the trial (Lim and In, 2019). Accordingly, in our study, the population was divided into several blocks based on demographic factors such as the participants` age. In such a context, the allocation process was concealed to prevent selection bias. To achieve this, the study population was initially anonymized by encoding participants with numerical identifiers. This encoding was performed by two of the study authors who did not participate in any data collection or analysis phases of the study. The concealment and anonymization process facilitated the blinding of data assessors. Consequently, the assessors were blinded to minimize the risk of bias during the data analysis phase.

In the control group, participants did not receive any intervention and were also asked if they had watched any movie leading to their anxiety or fear of heights in the recent past to decrease level of bias in the study data, allowing for the assessment of anxiety and fear of heights levels in individuals who were not subjected to the study intervention, thereby enabling comparison with the exposure group. This approach employs a nontreatment contrast, wherein the control group receives no intervention while the treatment group undergoes the active intervention, such as VRET, as in the case of this study (Hróbjartsson and Gøtzsche, 2003; Lin et al., 2012). On the other hand, in the intervention group, students were exposed to high altitude for eight 30-min sessions over 4 weeks by watching eight panoramic virtual reality videos. This 4-week (1 month) follow-up period was selected as it was considered appropriate according to evidence from the previous literature (Freeman et al., 2018). The videos included four real films and four animations. Participants mounted the VR headset (Samsung Gear connected to the Samsung S8 Live demo), and by playing each of these videos in a 360-degree, 3D space, they were immersed in high-altitude environments simulating various scenarios in a stepwise progression, including standing on skyscraper balconies, viewing cityscapes from elevated flyovers, experiencing parachute jumps and observing aerial drone footage near cliff edges. (Figure 1).

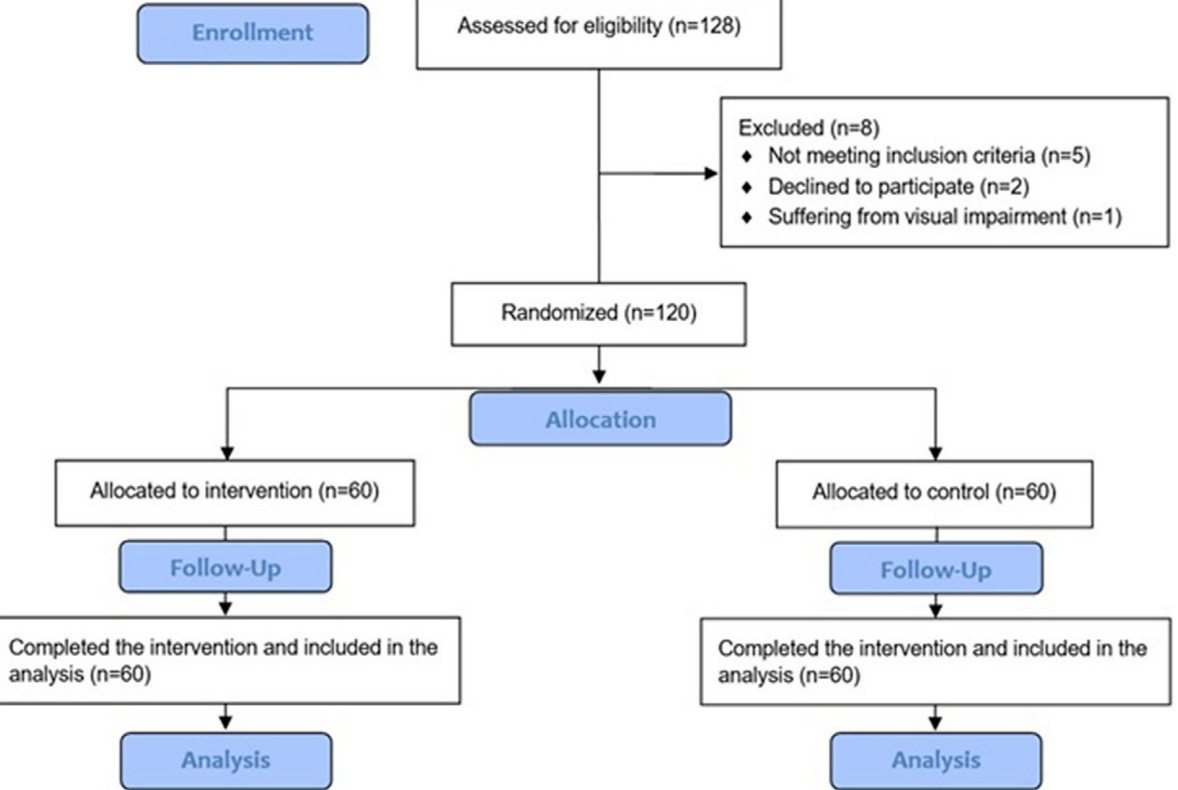

**Figure 1.** Flow diagram of the participant recruitment (CONSORT flow diagram).

The Samsung Gear VR constituted a lightweight, cordless virtual-reality headset that harnessed the Samsung Galaxy S8 smartphone as its primary display and processing unit, integrating seamlessly via the device's USB-C port. Equipped with 42 mm lenses that afforded a 101-degree field of view, it delivered fully immersive 360-degree visuals, while an ergonomic design and adjustable head strap ensured secure, comfortable wear during prolonged sessions. Powered by the Galaxy S8's 5.8-in. Quad HD+ Super AMOLED screen ($2,960 \times 1,440$ resolution), octa-core processor (Qualcomm Snapdragon 835 or Samsung Exynos 8,895) and 4 GB of RAM, the system achieved low-latency motion tracking through embedded accelerometers and gyroscopes that precisely mirrored head movements in virtual space.

As the main aim of this research was to investigate the role of psychological immersion in a VR environment on reducing acrophobia, we provided only virtual high spaces. To practice and exercise altitude in VR-based spaces, four steps were considered. Each step was practiced during two sessions. In one session VR movie and in the other session VR animation was applied. The participants experienced being at a gradually increasing altitude, starting at 10–20 m and progressing to a height of 50–60 m, 100–200 m and finally more than 500 m, respectively.

The simulation in VR created an environment for participants to become disconnected from their physical surroundings, leaving them immersed in the scenes, especially the sense of presence at height. As a result, the simulation permitted similar emotions to those that would occur in the real world. During the 4 weeks of exposure, participants were prohibited from any exposure to height outside of the scheduled sessions, as well as the use of alcohol, and were advised to have normal sleep. If a participant experienced cybersickness symptoms, such as nausea, dizziness or discomfort, the VR exposure would be halted immediately, and a therapist would intervene to ensure the participant's safety. Immediately after the sessions and at 1 month after, the questionnaires were given to all participants in the intervention and control groups to complete. To ensure the seamless and efficacious execution of the study intervention, a rigorously structured standardized exposure progression protocol comprised of the following items was meticulously formulated:

- Commencing intervention with low-altitude virtual environments designed to elicit only mild anxiety (e.g., distant horizon views or low balconies observed from a safe remove).
- Incrementally escalating both the perceived height and the dynamic complexity of scenarios across successive sessions, advancing methodically from stationary vantage points to active simulations such as aerial flyovers or controlled parachute descents.
- Requiring participants to self-report subjective anxiety within a pre-defined tolerable bandwidth at each stage.
- Authorize progression to the subsequent exposure tier exclusively upon confirmed habituation, defined as sustained anxiety reduction to manageable levels.

Moreover, several measures were implemented to ensure both the quality of the study intervention and the integrity of the collected data, including:

- Calibration and routine testing of the VR hardware to maintain accurate motion tracking and high-quality video playback.
- Verification of video content fidelity and consistency across all sessions to standardize the exposure.
- The involvement of experienced study authors in conducting the intervention, monitoring participants, collecting data and

enforcing safety protocols. In this regard, all authors underwent collective training sessions to adequately prepare for the study implementation across its various phases.
- Utilization of checklists and session logs to document strict adherence to intervention protocols.
- Continuous monitoring for cybersickness symptoms to promptly identify and manage any adverse effects.
- Comprehensive data auditing for completeness and accuracy, supplemented by periodic blinded reviews to safeguard against bias in data assessment.

### Beck anxiety inventory (BAI) questionnaire

The BAI is a standard questionnaire designed to measure the severity of anxiety in adolescents and adults (Appendix 1). This questionnaire has been reported to have a good reliability ($r = 0.72$, $p < 0.001$) and a very good validity ($r = 0.83$, $p < 0.001$) in the Iranian society (Kaviani and Mousavi, 2008). The Beck Anxiety Inventory (BAI) consisted of 21 items, each requiring the respondent to select one of four options indicating the severity of their anxiety symptoms. The scale ranged from 0 to 3 points per item. Each item described one of the most common symptoms of anxiety, encompassing mental, physical and panic-related manifestations. Consequently, the total BAI score ranged from 0 to 63 and was categorized as follows: 0–7 indicated no or minimal anxiety, 8–15 indicated mild anxiety, 16–25 indicated moderate anxiety and 26–63 indicated severe anxiety.

### Acrophobia questionnaire

The questionnaire included 14 items and was designed to assess the level of irrational fear of heights in individuals. It was scored using a 5-point Likert scale. To obtain the overall score, the scores of all items were summed. The maximum possible score was 70. The scoring scale was as follows: less than 24 indicated mild or no acrophobia, 24–50 indicated moderate acrophobia and 50–70 indicated severe acrophobia.

The validation process of the questionnaire comprised multiple stages, including assessment of face validity, content validity and construct validity. Finally, the reliability of the questionnaire was evaluated to ensure consistency and dependability of the measurement. The validity of the researcher-developed questionnaire was evaluated and confirmed based on the opinions of 10 psychiatrists and clinical psychologists. In this regard, the content validity was found to be within acceptable norms, with the Content Validity Ratio (CVR) exceeding the threshold of 0.62 based on Lawshe's table and the Content Validity Index (CVI), specifically the item-level CVI (I-CVI), surpassing 0.78. Its reliability was also assessed using Cronbach's alpha coefficient, which was above 0.72, indicating acceptable reliability (Gliem and Gliem, 2003). Finally, the participants were evaluated by a psychologist and diagnosed with acrophobia according to the DSM-5 criteria.

### Data analysis

Data were analyzed using SPSS, version 23. Descriptive statistics, Chi-square analysis tests, Independent t-test, repeated measures analysis and Bonferroni post hoc test at a significance level of 0.05 were reported. In this regard, the two-sample independent t-test was employed to compare the overall mean anxiety and fear of heights scores between distinct control and exposure groups. This parametric test evaluates whether the difference in group means is

statistically significant, utilizing a t-statistic and corresponding p-value, under the assumptions of normality, homogeneity of variances and independence of groups (Snedecor and Cochran, 1989). Additionally, a repeated measures ANOVA was conducted to assess changes in anxiety within the exposure group across three time points: pre-exposure, post-exposure and 1 month post-exposure (Greenwald, 1976; Keppel and Zedeck, 1989). The data analysis was performed by one researcher and subsequently validated by a second researcher.

## Results

As demonstrated in Figure 1, a total of 120 participants in two groups of VR exposure (N = 60 and the control group (N = 60) were enrolled in the study. In the intervention group, 23 (38.3%) were female. In the control group, 31 patients (51.7%) were female. The chi-square test presented that the distribution of genders in the two groups had no significant difference (p = 0.20). The mean ± SD age in the two groups was 22.20 ± 1.70 and 21.83 ± 3.24 years, respectively (Table 1). During the study intervention, no participant experienced cybersickness severe enough to warrant withdrawal from the study. Although five individuals who applied for the study did not meet the inclusion criteria, two opted not to participate, and one individual was excluded due to visual impairment, which precluded participation in the visually based virtual reality intervention.

The results of the independent t-test presented that the mean ± SD age did not differ significantly between the two groups (p = 0.44). The results of the Kolmogorov–Smirnov test presented that the variables of anxiety and acrophobia had a normal distribution before, after and 1 month after exposure in the two groups(P > 0.05). The total mean ±SD score of anxiety and acrophobia in both groups is presented in Table 2.

The mean ± SD score of anxiety in the intervention group before and after exposure was 15.92 ± 3.70 and 12.03 ± 2.54, respectively (Table 3). Moreover, Mauchly's test of sphericity was presented to be at an acceptable level (p = 0.073) (Moulton, 2010).

**Table 1.** Demographic characteristics of participants in the intervention and control groups

| Variable | Intervention (VR) (n = 60) | Control (n = 60) |
|---|---|---|
| Gender (Female/Male) | 23 (38.3%)/37 (61.7%) | 31 (51.7%)/29 (48.3%) |
| Age (years), Mean ± SD | 22.20 ± 1.70 | 21.83 ± 3.24 |

**Table 2.** Comparison of the total mean score of anxiety and fear of heights in two groups of exposure to virtual reality and control

| Group variable | Exposure group | Control group | Two-sample independent T-test |
|---|---|---|---|
| | Mean ± Standard deviation | Mean ± Standard deviation | |
| Total anxiety score | 14.01 ± 2.71 | 15.64 ± 4.31 | P = 0.034, t = 2.42 |
| Total fear of heights score | 27.12 ± 3.17 | 33.65 ± 8.2 | P = 0.026, t = 3.75 |

The repeated measure analysis presented that the mean scores of anxiety in the intervention and control groups at the selected timeframes were significantly different (p < 0.001). The results of the Bonferroni post hoc test presented that in the intervention group, the mean score of anxiety before exposure was significantly higher than after, and 1 month after exposure (p < 0.001). The independent t-test presented that the mean score of Post-exposure anxiety in the control group was significantly higher than the virtual reality exposure group (p = 0.03), but there was no significant difference before and 1 month after exposure (p = 0.12). The corresponding effect size (Cohen's d = 0.82) represents a large effect, indicating that VR exposure substantially reduced anxiety relative to the control condition.

Error Bars are portrayed in Figure 2 (a and b) for both groups before, after and 1 month after exposure based on the two studied factors. The mean ± SD scores of acrophobia in the intervention group were 32.43 ± 13.75 and 26.67 ± 10.22, before and after exposure, respectively (Table 4). The repeated measures analysis presented that the mean scores of acrophobia in the intervention and control groups at the selected timeframes were significantly different (p < 0.001). The results of the Bonferroni post hoc test presented that in the intervention group, the mean score of acrophobia before exposure was significantly higher than after exposure and 1 month later (p < 0.001). The independent t-test presented that the mean score of acrophobia immediately after and 1 month after exposure in the control group was significantly higher than the group exposed to virtual reality (p = 0.001), but scores in the pretest were not significantly different (p = 0.49). In this regard, the overall effect size of Cohen's d = 0.74 reflected a large effect.

## Discussion

As presented earlier, this study aimed to investigate the efficacy of utilizing VR technology on anxiety caused by acrophobia among the students of Birjand University of Medical Sciences in 2020. The findings presented that post-exposure anxiety was significantly higher in the control group than in the virtual reality group, with no significant differences before or 1 month after exposure. Similarly, acrophobia scores immediately and 1 month post-exposure were significantly higher in the control group, while pretest scores did not differ significantly. These suggest that virtual reality exposure was more effective than the control condition in reducing anxiety and acrophobia immediately after the intervention. While the reduction in anxiety was not sustained after 1 month, the decrease in acrophobia persisted, indicating a longer-lasting effect on fear reduction. In this regard, for acrophobia, virtual reality interventions have shown clinically significant improvements sustained at a 2-month follow-up, with certain literature reporting beneficial effects persisting for 6 months (Krijn et al., 2004; Francová et al., 2025). Overall, the findings suggested that VR exposure training led to meaningful and sustained reductions in both anxiety and fear of heights, with large effect sizes. The magnitude of these effects indicated that VR-based interventions were considerably more effective than the control condition in reducing emotional distress associated with height-related stimuli.

According to recent systematic reviews and meta-analyses in the existing literature, VR has been identified as an effective treatment modality for anxiety and phobia disorders. These studies indicate that VR interventions can alleviate the symptoms of anxiety disorders; however, they do not consistently demonstrate superiority

**Table 3.** Comparison of mean anxiety in two groups of exposure to virtual reality and control before, after and 1 month after exposure

| Variable | Group | One month after exposure Standard deviation ± Mean | Post-exposure Standard deviation ± Mean | Pre-exposure Standard deviation ± Mean | Repeated measure test |
|---|---|---|---|---|---|
| Anxiety | Exposure | 13.82 ± 2.86 | 12.03 ± 2.54 | 15.92 ± 3.70 | F = 35.23 P < 0.001 |
| | Control | 15.48 ± 5.69 | 15.55 ± 5.51 | 14.12 ± 5.71 | F = 45.66 P < 0.001 |
| | | P = 0.29 T = 1.05 | P = 0.03 T = 2.13 | P = 0.35 T = 0.94 | Independent T-test |
| | | | 0.8204 | | Cohen's d effect size |

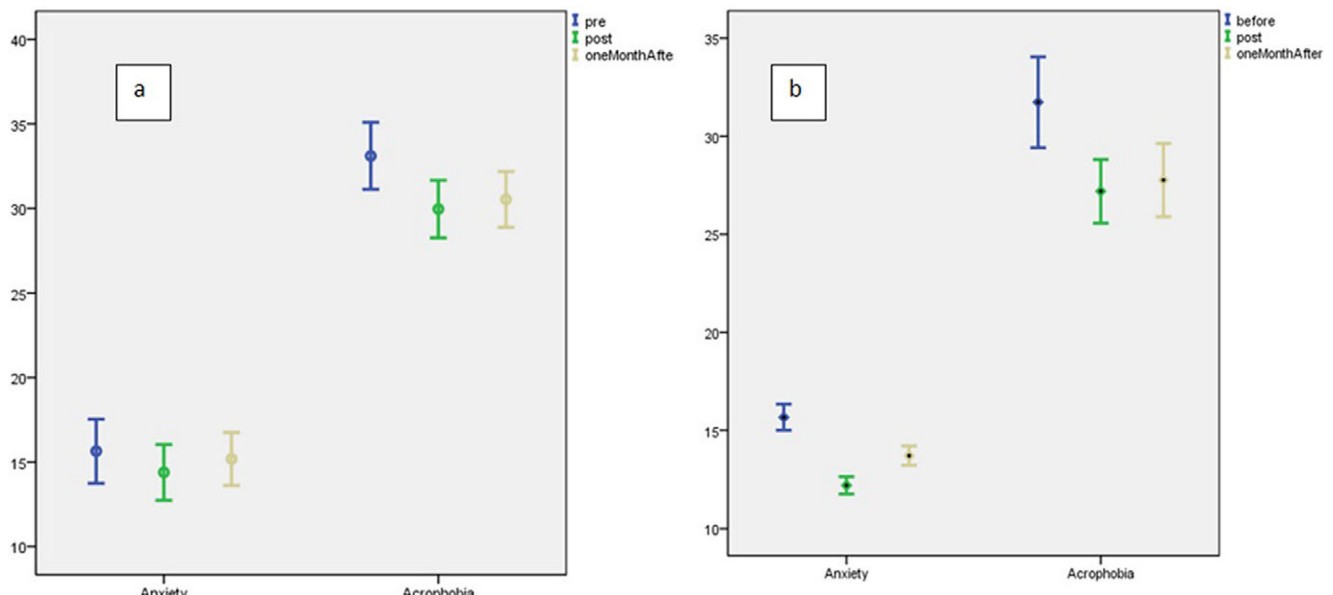

**Figure 2.** a: Error Bar in the control group before, after and 1 month after exposure based on two studied factors. b: Error Bar in the group of exposure to virtual reality before, after and 1 month after exposure based on two studied factors.

**Table 4.** Comparison of the average fear of heights in the two groups of exposure to virtual reality and control before, after and 1 month after exposure

| Variable | Group | One month after exposure Standard deviation ± Mean | Post-exposure Standard deviation ± Mean | Pre-exposure Standard deviation ± Mean | Repeated measure test |
|---|---|---|---|---|---|
| Fear of heights | Exposure | 27.65 ± 9.47 | 26.67 ± 10.22 | 32.43 ± 13.75 | F = 57.54 P < 0.001 |
| | Control | 33.22 ± 8.07 | 33.25 ± 7.22 | 33.82 ± 7.17 | F = 0.94 P = 0.38 |
| | | P = 0.001 T = 2.94 | P = 0.001 T = 3.34 | P < 0.001 T = 4.07 | Independent T-test |
| | | | 0.7436 | | Cohen's d effect size |

over traditional therapeutic approaches (Zeng et al., 2025). Furthermore, the degree of effectiveness varies across different types of anxiety disorders (Wong et al., 2023). Given the limited scope and heterogeneity of the available research, it remains challenging to determine the most effective therapeutic approach. This underscores the importance of conducting original empirical studies to provide additional insights and strengthen the existing body of knowledge on this topic (Kuleli et al., 2025).

The findings of this study were consistent with the rest of the studies within the Iranian literature. In this regard, in line with our study findings, the effectiveness of using diverse VR environments for the treatment of acrophobia and darkness phobia was presented in the recovery of these patients (Jafari and Safae, 2020). Similarly, another study reported that the treatment of virtual reality has a significant effect on reducing fear of flying in people with aviophobia (fear of flying) (Eslami et al., 2013). It is found that the

therapeutic effects of using VR simulations can be even continued in the long term, which is in line with our findings (Eslami et al., 2013). Moreover, virtual reality exposure therapy has been presented to significantly alleviate symptoms of acrophobia and anxiety sensitivity in female adolescents (Azimisefat et al., 2022).

In the global context, in line with our study findings, it is presented that applying VR behavioral-cognitive software is effective in reducing acrophobic symptoms immediately after treatment and at 3-month follow-up (Donker et al., 2019). A pilot study presented that VR is an effective tool in treating acrophobia (Levy et al., 2016). Similarly, a clinical trial which examined the effectiveness and efficiency of exposure therapy in a VR setting compared to real life in the treatment of acrophobia, presented significant improvements in anxiety, avoidance and behavioral scales (Coelho et al., 2008). Moreover, a review study, which discussed the dissemination of VR into clinical practice and what VR will offer physicians in the future, presented that such dissemination will happen not only in an adult population but also for younger patients. It was presented that VR has a special appeal for young people as it links simply with its playful elements to this population and might offer treatment or at least preventive interventions (Amiri et al., 2019). Overall, the literature indicates that the impact of VR on mental health disorders does not significantly differ between low-resource and high-resource settings. In this context, as presented earlier in this study, all studies within the literature have demonstrated the positive effects of utilizing VRET.

Although virtual reality has demonstrated considerable effectiveness in mental health services, several challenges hinder its widespread adoption. A key concern is whether therapeutic gains achieved in VR translate effectively to real-world settings and how to measure this transfer (López Del Hoyo et al., 2024). Additionally, some users experience adverse effects such as cybersickness, fatigue, symptom exacerbation and physical discomfort (Bell et al., 2020; López Del Hoyo et al., 2024). Furthermore, VR therapy is not suitable for all patients, particularly those with conditions like severe motion sickness, epilepsy or certain psychiatric disorders, necessitating careful clinical evaluation (Kothgassner et al., 2023). Importantly, the requirement for specialized equipment and technical expertise limits accessibility, especially in low-resource settings, where cost and infrastructure pose significant barriers (Srivastava et al., 2014; Bell et al., 2020).

## Strengths and limitations

The study has a limitation that should be acknowledged. Specifically, the intervention was conducted in 2020, a period during which virtual reality technology was less advanced and had certain limitations. Moreover, the follow-up period of the study was limited to 1 month due to constraints in time and resources. Future research may extend this duration to obtain more precise and comprehensive insights. Furthermore, the study sample was restricted to students from a single university due to limited resources, which represents another limitation of the study. This was considered to restrict the generalizability of the findings to broader populations that may exhibit different baseline anxiety patterns, treatment expectations or levels of familiarity with virtual reality. This limitation may be addressed in future research by including more diverse and representative samples. Moreover, yet another limitation of the study involved the use of the nontreatment contrast approach, wherein the control group received no intervention.

The study has some strengths to address. In this regard, few studies have used VR headsets for exposure therapy in low-resource settings, and the present study is one of the few studies that have used cutting-edge technology for exposure within the context. This may be considered a distinctive characteristic of our study, enhancing its contribution beyond mere replication in high-resource settings. The high sample size of the present study is another strength of this trial. Moreover, as VR has various applications in healthcare, especially in the field of psychiatric care and treatment, mental healthcare providers will be able to use this technology in a more evidence-based approach in treating patients with mental health disorders, particularly with acrophobia.

## Conclusions

The study concluded that virtual reality technology constitutes an effective intervention for reducing anxiety and acrophobia. This approach demonstrates considerable potential as a therapeutic modality for psychiatrists managing patients with acrophobia. More clinical trials, however, are needed to assess the feasibility, therapeutic effects and mechanisms of online applications.

**Open peer review.** To view the open peer review materials for this article, please visit http://doi.org/10.1017/gmh.2026.10160.

**Supplementary material.** The supplementary material for this article can be found at http://doi.org/10.1017/gmh.2026.10160.

**Data availability statement.** The study data will be available upon a reasonable request from the corresponding author.

**Acknowledgments.** The authors acknowledge the contribution from the Dentistry Clinical Research Development Center of Birjand University of Medical Sciences in consultation during the research. The authors also used ChatGPT-4 in order to rewrite the text of the manuscript to ensure correct grammar and wording.

**Author contribution.** MZB, SRS and CB conceptualized and wrote the original draft; FO and SFMP analyzed the data, and SM and SBM collected the data. MK revised the final version of the manuscript.

**Financial statement.** This research was funded by Birjand University of Medical Sciences, Birjand, Iran (grant no.: 455409).

**Competing interest.** There is no conflict of interest within the research.

**Ethics statement.** The present study was granted by Birjand University of Medical Sciences and approved by the corresponding ethics committee (IR.BUMS.REC.1399.088). The study was also registered in the Iranian Registry of Clinical Trials (IRCT20200619047830N1). It is accessible via the following link: https://irct.behdasht.gov.ir/. Moreover, Informed consent was taken from all of the study participants during the research.

**Consent for publication.** Not applicable.

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
