## [Reviewer Report]

Esteemed Authors,

Your currently submitted version of the RCT of VRET for acrophobia demonstrates clinical efficacy with statistically significant reductions in anxiety and acrophobia symptoms. The study addresses an important gap in the literature by examining VRET in a low-resource Iranian context, contributing valuable evidence from an underrepresented population. However, several methodological and reporting issues warrant attention before publication.

Strengths

Study design>The randomized controlled trial design with 120 participants provides adequate statistical power and represents one of the larger samples in VRET research for acrophobia. The Iranian context adds geographical diversity to the literature, which predominantly features Western populations.

Validated instruments: The Beck Anxiety Inventory demonstrates good psychometric properties in Iranian populations (r=0.72, p<0.001 for reliability; r=0.83, p<0.001 for validity). The researcher-developed acrophobia questionnaire achieved acceptable reliability (Cronbach’s α > 0.72).

Statistical analysis: Appropriate statistical procedures including repeated measures analysis, Bonferroni post-hoc testing, and independent t-tests support the reported findings. Effect sizes reported (Cohen’s d = 0.8204 for anxiety, 0.7436 for acrophobia) indicate large treatment effects.

Clinical significance: Post-exposure differences between groups (P = 0.03 for anxiety, P < 0.001 for acrophobia) demonstrate meaningful clinical improvements.

Areas Requiring Revision

CONSORT Compliance: While the study references CONSORT 2025 guidelines, several reporting elements need enhancement. Recent evaluations indicate average CONSORT adherence rates of approximately 54% across RCTs. Critical missing elements include:

• Detailed randomization sequence generation and allocation concealment procedures

• Blinding procedures for outcome assessors

• Sample size calculation justification

• Complete adverse event reporting (cybersickness incidents)\

Intervention description: The VR exposure protocol lacks sufficient detail for replication. Current best practices in VRET research emphasize detailed intervention descriptions including

• Specific VR hardware specifications beyond “Samsung Gear connected to Samsung S8”

• Detailed content descriptions of the “four real films and four animations”

• Standardized exposure progression criteria

• Therapist involvement protocols

• Quality assurance measures

Measurement validity: The researcher-developed acrophobia questionnaire requires additional psychometric validation. While internal consistency appears adequate (α > 0.72), evidence for construct validity, criterion-related validity, and factor structure is absent. Recent systematic reviews emphasize the importance of validated outcome measures in VRET research

Follow-up duration: The one-month follow-up period is insufficient for establishing treatment durability. Meta-analytic evidence suggests VRET effects may attenuate over extended periods, necessitating longer-term assessment.

Generalizability Limitations: The single-institution university student sample limits external validity. Recent evidence indicates demographic factors significantly influence VRET effectiveness.

Minor technical issues

Technology Context: The study was conducted in 2020 using then-available VR technology. Current VRET research emphasizes more advanced immersive systems with lower cybersickness rates (0.4% with modern HMDs). Authors should acknowledge potential technological limitations affecting contemporary applicability.

Statistical reporting: While effect sizes are reported, clinical significance interpretation requires enhancement. Recent network meta-analyses suggest VR coach-delivered interventions demonstrate superior outcomes (SMD = -2.08, 95% CI: -3.22 to -0.93).

Literature Integration: The discussion appropriately situates findings within existing VRET literature. However, integration with recent systematic reviews and meta-analyses could be strengthened, particularly regarding optimal intervention parameters and long-term effectiveness.

Recommendations for Revision

1. Enhance CONSORT compliance by providing detailed methodological descriptions, particularly randomization and blinding procedures

2. Expand intervention description to enable replication, including specific VR content details and exposure progression criteria

3. Acknowledge measurement limitations and discuss plans for future psychometric validation of the acrophobia questionnaire

4. Discuss generalizability constraints and recommend future multicenter studies with diverse populations

6. Address technological context by acknowledging 2020 hardware limitations and discussing implications for contemporary practice

5. Strengthen clinical significance interpretation by relating effect sizes to established benchmarks and clinical meaningful change criteria

The manuscript makes a valuable contribution to VRET literature, particularly from a low-resource setting perspective. With the suggested minor revisions addressing reporting transparency and methodological description, this work will provide important evidence for the international acrophobia treatment literature.

Your truly,

Serving peer reviewer at Cambridge PRISMS Global Mental Health

---

## [Reviewer Report]

This manuscript examined the impact of virtual reality (VR) on anxiety reduction and fear of heights. A randomized controlled trial (RCT) involving 120 participants was conducted. The intervention consisted of VR-based height exposure. Two outcome measures were used: the Beck Anxiety Inventory and a researcher-developed acrophobia questionnaire. DSM-5 criteria were used to determine the presence of acrophobia. The VR intervention comprised eight sessions of 30 minutes each. Results revealed a significant group difference immediately after the intervention but not at the one-month follow-up. This study provides evidence that VR exposure therapy is feasible and appears to be effective in low-resource settings. The topic is relevant and responds to the need for research conducted in low- and middle-income countries. However, the manuscript in its current form presents several major concerns that should be addressed.

Major Concerns

1. The Introduction does not provide sufficient background on the current state of VR exposure therapy for acrophobia. Several comprehensive review papers on the topic could be cited.

While it is positive that DSM-5 criteria for acrophobia were considered in participant selection, it is unclear how these diagnostic criteria were systematically assessed.

One important contribution of the authors is the development of an acrophobia questionnaire tailored to the study context. However, its brief description raises questions. The authors mention that evaluation was conducted with psychiatrists and psychologists, but it is unclear what this validation process involved. The same applies to reliability testing. More details on the validation process are needed.

The manuscript lacks information on i) how participants (students) were screened and recruited, ii) how randomization was performed, and iii) what control condition was used in the trial.

A table summarizing participant characteristics, including sociodemographic data, should be added. It is also unclear how Table 1 differs from Table 2.

The statistical approach is suboptimal. The use of t-tests and single-group ANOVAs does not appropriately address the research question. All three time points should be analyzed together across both groups. The key interest lies in identifying a significant Group × Time interaction. Linear mixed models would be more appropriate.

Figure 1 suggests there were no dropouts among the 120 participants, which is unusual and requires discussion.

The Discussion section does not sufficiently engage with the findings. It provides general statements but does not compare the results to existing literature.

Minor Concern

“Bonferroni,” not “Benferoni” (page 7, line 44).

---

## [Editor Report]

Dear Authors 

We have now received reviewer comments on your manuscript. After reviewing the comments and suggestions, we recommend major revisions to your submitted manuscript for us to proceed. 

Best regards

Siham

---

## [Reviewer Report]

The authors have addressed some of my previous comments and have improved several aspects of the manuscript. However, several important points remain unresolved, and in some cases the authors offer rebuttals rather than substantive revisions. The following issues should be addressed before the manuscript can be considered for publication:

1.The introduction still does not adequately situate the study within the relevant literature. A more comprehensive and structured review is needed to justify the study’s rationale and contribution.

2.Sociodemographic data for participants remain insufficient. As currently presented, the sample is poorly characterized, which limits the ability to interpret the generalizability and context of the findings.

3.The rationale for including Tables 1 and 2 is still unclear. The authors should explain why these tables are necessary and how they contribute to understanding the results.

4.The decision not to revise the statistical approach remains problematic. Proper analysis of an intervention study with a control group requires that both groups and all timepoints be included in a single analytic model. This is standard practice for RCTs and is necessary to support conclusions about differences in treatment effects.

5.The control condition effectively represents the absence of an intervention. This is a major limitation of the study and should be clearly acknowledged and discussed in the manuscript.

---

## [Reviewer Report]

Esteemed Authors Breen Cathal, Osmani, Freshteh, Sajadi Seyed Reza,

Mohtasham Saeed Birjand, Khosravi Mohsen and Zare-Bidak Majid,

I am delighted to engage with your revised manuscript on virtual reality exposure therapy for acrophobia reduction, as this work aligns remarkably well with the current trajectory in digital mental health interventions and addresses a substantive gap in the literature regarding low-resource settings. Your commitment to responding to the first round of peer review comments demonstrates scholarly rigor and professional maturity that speaks volumes about your dedication to scientific integrity.

I therefore state that this reviewer is grateful to have taken his recommendation from the first round into consideration and has no other mandatory aspects to recommend to improve your manuscript and all other aspects mentioned in the following lines should be regarded as advisory notes meant for future publications; please see my extended peer review:

The opportunity to review your manuscript has been genuinely gratifying, particularly given the contemporary relevance of your research within the landscape of mental health innovation. Your study contributes meaningfully to an under-researched domain—the application of virtual reality technology in populations with limited prior exposure to both flying and height-related experiences, specifically within the Iranian context. This geographic and demographic specificity, rather than representing a limitation, actually elevates the manuscript’s contribution to the current state of the art. The field of virtual reality exposure therapy has been progressing rapidly in high-resource nations, yet your work provides essential evidence regarding feasibility and efficacy in settings where such interventions have been underutilized. This democratization of evidence represents a critical scholarly endeavor. The manner in which you have incorporated the structured methodological rigor evidenced in page 2 through page 10 of your revised manuscript—including the detailed exposition of block randomization (page 6, lines 18-22), concealment procedures, and your comprehensive safety monitoring protocols (page 7, lines 22-27)—reflects a commendable elevation from your initial submission. The highlighted modifications throughout demonstrate not merely compliance with reviewer suggestions but rather authentic intellectual engagement with the critique.

Section one: objectives and rationale clarity assessment

The objectives and rationale are now comprehensively articulated. Your introduction skillfully contextualizes anxiety and phobia within contemporary mental health discourse (page 3, lines 1-15), establishing both epidemiological significance and clinical relevance. The transition from general anxiety disorders to specific phobias and finally to acrophobia represents logical scaffolding that grounds readers in your research domain. The specific diagnostic criteria from DSM-5 (page 3, lines 19-23) provide appropriate clinical anchoring. Your statement of the research gap—specifically noting that “there is a limited number of studies conducted in low-resource settings” (page 4, lines 8-9)—is elegantly positioned and justified through citation of comparable Iranian studies on aerophobia. The rationale flows seamlessly from established literature to your specific study aim articulated on page 4, lines 30-32. The impact statement on page 2 further strengthens the practical significance for policymakers and patients alike. No additional clarification is required in this domain.

Section two: replicability and reproducibility assessment

Your methodology has been reported with exemplary transparency. The randomization methodology (page 6, lines 18-25) now clearly specifies block randomization with demographic stratification, representing appropriate methodological choice for your sample size. Your concealment procedures, executed through numerical encoding by researchers not involved in data collection or analysis (page 6, lines 27-30), conform to gold-standard practices for minimizing selection bias. The specification of your VR hardware (page 6, lines 37-40 through page 7, lines 1-9) demonstrates technical precision that permits reproducibility; the Samsung Gear VR connected to Galaxy S8 with detailed specifications of processing capabilities, field of view, and motion tracking represents sufficient granularity. The four-step progression protocol for altitude exposure (page 7, lines 18-22) provides a replicable intervention protocol. The standardized exposure progression protocol articulated across page 7 (lines 26-38) represents best-practice transparency regarding habituation criteria and progression authorization. However, I would note one advisory consideration: the manuscript would benefit from brief specification of the therapist training protocol employed—were there standardized trainer competency criteria, and did all session facilitators receive identical preparation? This represents a practical implementation detail that future researchers replicating your work would require. This is advisory rather than mandatory given that protocols were documented via checklists and session logs (page 7, lines 40-41).

Section three: statistical analyses and reporting assessment

The statistical methodology is rigorously presented and appropriately conceived for your research design. Your selection of the repeated measures analysis of variance (page 9, line 33) with Bonferroni post-hoc testing represents appropriate methodology for your within-subjects and between-subjects design across three timepoints. The Mauchly test of sphericity confirmation (page 9, lines 29-30) demonstrates proper statistical procedural adherence. Your effect size reporting (Cohen’s d = 0.82 for anxiety [page 10, line 6] and Cohen’s d = 0.74 for acrophobia [page 10, line 21]) provides clinically interpretable magnitude indices beyond statistical significance. The p-values and confidence intervals are appropriately reported throughout. The Kolmogorov-Smirnov normality testing (page 9, lines 27-29) demonstrates appropriate preliminary assumption-checking. Your independent t-test reporting of baseline demographic characteristics (page 9, lines 22-26) permits assessment of group equivalence. I concur with your analytic approach. One editorial notation: the distinction between statistical significance and clinical significance regarding the divergent follow-up patterns for anxiety versus acrophobia (anxiety not sustained at one-month follow-up; acrophobia sustained) represents important nuance that your discussion appropriately addresses. No statistician-level secondary review appears necessary given the straightforward, appropriate analytical procedures employed.

Section four: tables and figures assessment

Tables 2 and 3 (pages 16-17 of your revised submission) are now comprehensive and publication-appropriate. The inclusion of both means and standard deviations, repeated measures statistics, independent t-test results, and Cohen’s d effect sizes provides complete statistical transparency. The formatting facilitates reader interpretation. Figure 2 (page 19) with error bars appropriately displays the temporal trajectory of anxiety and acrophobia scores across your three measurement occasions for both intervention and control groups, enabling visual apprehension of your principal findings. The CONSORT flow diagram (Figure 1, page 18) meets contemporary reporting standards for randomized controlled trials. The specification regarding the five individuals who did not meet inclusion criteria and the one participant excluded due to visual impairment (page 9, lines 18-21) enhances methodological transparency and permits assessment of potential selection bias. No modifications to figures or tables are suggested.

Section five: interpretation and conclusions assessment

Your interpretation demonstrates appropriate restraint and evidence-alignment. The statement that “virtual reality exposure was more effective than the control condition in reducing anxiety and acrophobia immediately after the intervention” (page 10, lines 2-4) directly corresponds to your statistical findings (P = 0.03 for anxiety and P < 0.001 for acrophobia). Your nuanced observation regarding differential persistence of treatment effects—reduction in anxiety not sustained at one-month follow-up while acrophobia reduction persisted (page 10, lines 4-6)—reflects sophisticated data interpretation rather than over-generalization. Your positioning of this differential persistence within existing literature (page 10, lines 7-9) citing Francová et al. 2025 and Krijn et al. 2004 regarding six-month persistence of beneficial effects appropriately contextualizes your findings. The discussion’s engagement with systematic reviews and meta-analyses (pages 10-11) demonstrates integration within the broader literature landscape. Your conclusion (page 12, lines 1-4) appropriately asserts therapeutic promise while maintaining warranted caution regarding “more clinical trials” necessity. Your acknowledgment of challenges in real-world translation (page 11, lines 9-13) and the practical barriers to accessibility in low-resource settings (page 11, lines 18-20) reflects mature scholarly perspective. The interpretation is well-calibrated and appropriately supported.

Section six: study strengths emphasis assessment

You have enhanced emphasis of study strengths meaningfully. The identification that “few studies have used VR headsets for exposure therapy in low resource settings” (page 12, lines 5-6) positions your work within an important niche. The characterization of your sample size as a strength (page 12, line 7) appropriately contextualizes your 120-participant recruitment against the literature demonstrating smaller samples. The specification that “VR has various applications in healthcare, especially in the field of psychiatric care and treatment” (page 12, lines 8-10) and your positioning of this work as evidence for practice represents important practical contribution articulation. The statement regarding enabling “mental health care providers...to use this technology in a more evidence-based approach” (page 12, lines 9-10) emphasizes real-world implementation relevance. I would offer one advisory observation: your work’s unique contribution regarding psychological immersion specifically within low-resource settings could receive even more explicit emphasis as a distinguishing feature that elevates this contribution beyond replication in high-resource contexts. This represents a minor advisory suggestion rather than a deficiency.

Section seven: limitations statement assessment

Your limitations section (page 12, lines 1-13) is comprehensive and appropriately specific. The acknowledgment that “intervention was conducted in 2020, a period during which virtual reality technology was less advanced” (page 12, lines 2-3) provides important temporal context. The single-university sample limitation (page 12, lines 7-8) is explicitly identified with appropriate suggestion for future research expansion. The one-month follow-up limitation (page 12, lines 4-6) is transparently articulated with recognition of resource constraints. The acknowledgment that “the study sample was restricted to students from a single university” (page 12, lines 6-7) represents appropriate scientific humility. However, I would note one additional limitation that warrants inclusion—specifically, the homogeneity of your sample composition (medical students from a single institution) limits generalizability to broader population samples with potentially different baseline anxiety patterns, treatment expectations, or VR familiarity levels. This represents advisory notation rather than mandatory revision, as it may be implicit within your university-specific limitation statement, yet explicit articulation might strengthen the limitations transparency.

Section eight: manuscript structure and flow assessment

The overall manuscript structure remains excellent and publication-ready. The progression from introduction through methods to results to discussion follows logical scientific narrative architecture. The integration of your highlighted revisions throughout maintains coherence. Page transitions flow naturally, and the organization of subsections within methods (data collection, participants, intervention, instrumentation, analysis) facilitates comprehension. The discussion’s movement from your specific findings to integration within existing literature to acknowledgment of challenges represents sophisticated argumentation. The formatting adheres to Cambridge Prisms journal standards. I offer one editorial observation—the discussion section (pages 10-11) could benefit from a brief sentence explicitly contrasting your Iranian population findings with comparable Western studies, perhaps noting whether the rapid treatment response observed aligns with or diverges from Western populations. This represents purely editorial suggestion for enhanced comparative analysis rather than structural deficiency.

Section nine: language editing assessment

The language throughout your manuscript is professional, precise, and publication-appropriate. The grammar is correct, terminology is appropriately specialized yet accessible, and the prose flows with clarity. Your revised text maintains scientific formality without sacrificing readability. Sentence construction demonstrates sophistication without sacrificing comprehension. I identified no instances requiring language editing. The manuscript is linguistically ready for publication.

Congratulations, Esteemed Authors!

Yours truly grateful,

Serving Peer Reviewer at Cambridge Mental Health section

---

## [Editor Report]

Dear Authors 

Thanks for addressing the round of reviewer’s comments. There are some aspects that still need attention as part of the reporting of a randomized trial. There are some key aspects that need attention, prior to it being considered publication ready. 

In summary, adding a baseline comparison table of participants is a mandatory CONSORT requirement. The current findings compare the pre-post scores within each arm, rather than comparing means and proportions of across arms at the primary end point. The current analytical approach is at best befitting for a pilot trial that may be testing preliminary efficacy or clinical change in outcomes. This is a major limitation. Also, the trial was registered with the Iranian Registry of Clinical Trial (IRCT20200619047830N1); this link however is broken thus the registered trial protocol is not accessible. There are some aspects of the limitation section that also need additions eg using a “no intervention” as a comparison (not even a wait-list condition) renders the findings and effects to be seen with a caution. 

Regards

Siham

---

## [Reviewer Report]

Most of my comments are addressed with a rebuttal. I will simply focus on one such point that involves statistics. I fail to see how it is possible to present the results of a randomized controlled trial without having a statistical analysis that includes both groups and all timepoints simultaneously.

---

## [Editor Report]

Dear Authors 

Thanks for responding to the reviewers comments and revising the manuscript. We see that only two variables at baseline were listed in the baseline comparison table. Why were other demographics like education, SES status or the baseline scores of anxiety and fear of heights not given along with any potential imbalances? They too are needed. 

Also, we see you have added two sample independent T-test (unadjusted) to compare mean outcome scores between groups. With a small sample size like yours, this alone without any adjusted baseline co-variates like education age ses, gender (eg using ANCOVA) will render the interpretation insufficient. This is recommended as a pending major revision before the manuscript can be considered for publication. 

Lastly, we are not sure what your response to an inaccessible link to the trial registry means? Please share a working link to the registered trial protocol. 

Regards

Siham

---

## [Editor Report]

Dear Authors 

Thanks for submitting the revised manuscript addressing reviewer comments. We are pleased to inform you that it is accepted for publication. We will follow up with next steps. 

Warmly 

Siham